# A serpin gene from a parasitoid wasp disrupts host immunity and exhibits adaptive alternative splicing

**Zhichao Yan[1,2], Qi Fang[2], Jiqiang Song[2], Lei Yang[2], Shan Xiao[2], Jiale Wang[2], Gongyin Ye[2]\***

**1** Department of Entomology, Nanjing Agricultural University, Nanjing, China, **2** State Key Laboratory of Rice Biology & Ministry of Agricultural and Rural Affairs Key Laboratory of Molecular Biology of Crop Pathogens and Insects, Institute of Insect Sciences, Zhejiang University, Hangzhou, China

\* chu@zju.edu.cn

## Abstract

Alternative splicing (AS) is a major source of protein diversity in eukaryotes, but less is known about its evolution compared to gene duplication (GD). How AS and GD interact is also largely understudied. By constructing the evolutionary trajectory of the serpin gene *PpSerpin-1* (*Pteromalus puparum* serpin 1) in parasitoids and other insects, we found that both AS and GD jointly contribute to serpin protein diversity. These two processes are negatively correlated and show divergent features in both protein and regulatory sequences. Parasitoid wasps exhibit higher numbers of serpin protein/domains than nonparasitoids, resulting from more GD but less AS in parasitoids. The potential roles of AS and GD in the evolution of parasitoid host-effector genes are discussed. Furthermore, we find that *PpSerpin-1* shows an exon expansion of AS compared to other parasitoids, and that several isoforms are involved in the wasp immune response, have been recruited to both wasp venom and larval saliva, and suppress host immunity. Overall, our study provides an example of how a parasitoid serpin gene adapts to parasitism through AS, and sheds light on the differential features of AS and GD in the evolution of insect serpins and their associations with the parasitic life strategy.

**Data Availability Statement:** Alignments, trees and iTOL annotation files are publicly available on FigShare (https://doi.org/10.6084/m9.figshare.21545598.v1). The accession numbers of the raw

## Author summary

Alternative splicing (AS) and gene duplications (GD) significantly contribute to protein diversity in eukaryotes, yet their interplay and associations with life strategies remain largely understudied. Parasites recruit diverse virulence genes to counter host defenses, providing a unique window into the interplay between AS and GD. Investigating the evolution of the *PpSerpin-1* gene (*Pteromalus puparum* serpin 1) in parasitoids and other insects, we uncovered a negative correlation between AS and GD, distinct sequence features, and associations to the parasitic life strategy. We also demonstrate how AS enhances *PpSerpin-1* protein diversity, which participates in parasitoid wasp venom and larval saliva

proteomics data are PXD044136 and PXD044199 in PRIDE (Proteomics IDEtification Database). The numerical data used in all figures are included in S1 Data.

**Funding:** This work was supported by the Key Program of National Natural Science Foundation of China (NSFC) (grant no. 31830074 to G.Y.Y.), NSFC (grant no. 31701843 to Z.C.Y.), the Regional Joint Fund for Innovation and Development of NSFC (grant no. U21A20225 to G.Y.Y.), NSFC (grant no. 32072480 to Q.F. and no. 32001964 to L.Y.), Natural Science Foundation of Hainan Province (grant no. 323QN262 to Y.Z.C.), Natural Science Foundation of Zhejiang Province (grant No. LTGN23C140001 to F. Q.), and Major International (Regional) Joint Research Project of NSFC (grant no. 31620103915 to G.Y.Y.). The funders had no role in study design, data collection and analysis, decision to publish, or preparation of the manuscript.

**Competing interests:** All authors declare that there is no conflict of interest.

to regulate host immunity. To our knowledge, this also represents the first functional analysis of a larval salivary gene in parasitoid wasps.

## Introduction

Alternative splicing (AS) is a frequent regulatory process of transcription in animals, plants and fungi [1,2]. Through differential inclusion/exculsion of exons and introns, AS produces multiple variant proteins from a single gene [1,2]. AS and gene duplication (GD) are important sources of genetic innovation and protein diversity [3–6]. Compared to GD, relatively little is known about the evolution of AS and its role in adaptation [3,7,8]. Moreover, the relationship and difference between these two evolutionary processes, AS and GD, are largely understudied [4,9,10].

Pathogens and parasites recruit diverse virulence genes to overcome their hosts and adapt to coevolution with hosts [11–14]. For example, to ensure successful parasitism, parasitoid wasps often inject venom to manipulate their hosts' immunity, metabolism, development and even behavior [14,15]. Driven by frequent host shifts and the arms race between parasitoid wasps and their hosts, parasitoid venoms show rapid compositional turnover and high sequence evolutionary rates [14,16–18]. Evolutionary processes for parasitoid venom gene recruitment include GD [19–21], AS [22,23], lateral gene transfer [24,25], and single-copy gene co-option for venom functions [17,26].

Serpins (serine protease inhibitors) are a widely distributed protein superfamily in all kingdoms of life [27–29]. Serpins contain similar structures with three β-sheets, 7–9 α-helices and an exposed reactive center loop (RCL) on the C-terminus, which determines serpin activity and specificity [30,31]. Most serpins are inhibitors that irreversibly inhibit target enzymes by conformational change [32], where the hinge region of RCL acts as a "spring" and inserts into β-sheets [33,34]. Through their inhibitory activities, serpins play central roles in proteolytic cascades, e.g., coagulation and inflammation in mammals and the prophenoloxidase (PPO) casecade and Toll pathway in insects [28,29].

In parasitoid wasps, serpins are a common venom component and have been reported in numerous parasitoid venoms [17,20–23,35–43]. Both AS and GD have been reported in the recruitment of serpin into parasitoid venoms [21,22]. Previously, we isolated a venom serpin protein from *Pteromalus puparum*, a generalist parasitoid wasp that parasitizes pupae of several butterfly species [22]. This venom serpin is produced by the *PpSerpin-1* gene through AS and suppresses the host's melanization immunity [22]. In contrast, extensive GD of serpin was reported in the venom apparatus of a parasitoid wasp, *Microplitis mediator* [21]. Therefore, parasitoid venom serpins may provide a promising model for comparative studies on the evolution and ecological adaptation of AS and GD.

Here, we analyzed the evolutionary trajectory of *PpSerpin-1* and compared the different features of AS and GD in serpin evolution and their associations with the parasitic life strategy. In addition, we demonstrate that *PpSerpin-1* is involved in wasp's immune responses and has been recruited to both wasp venom and larval saliva, to regulate host immunity.

## Results

### AS contributes to serpin protein diversity with GD and domain duplication in insects

Utilizing the accumulated transcriptomic data, we found that the *PpSerpin-1* gene has alternative splicing, with two different N-terminal and 21 C-terminal AS forms (Figs 1A and S1). At

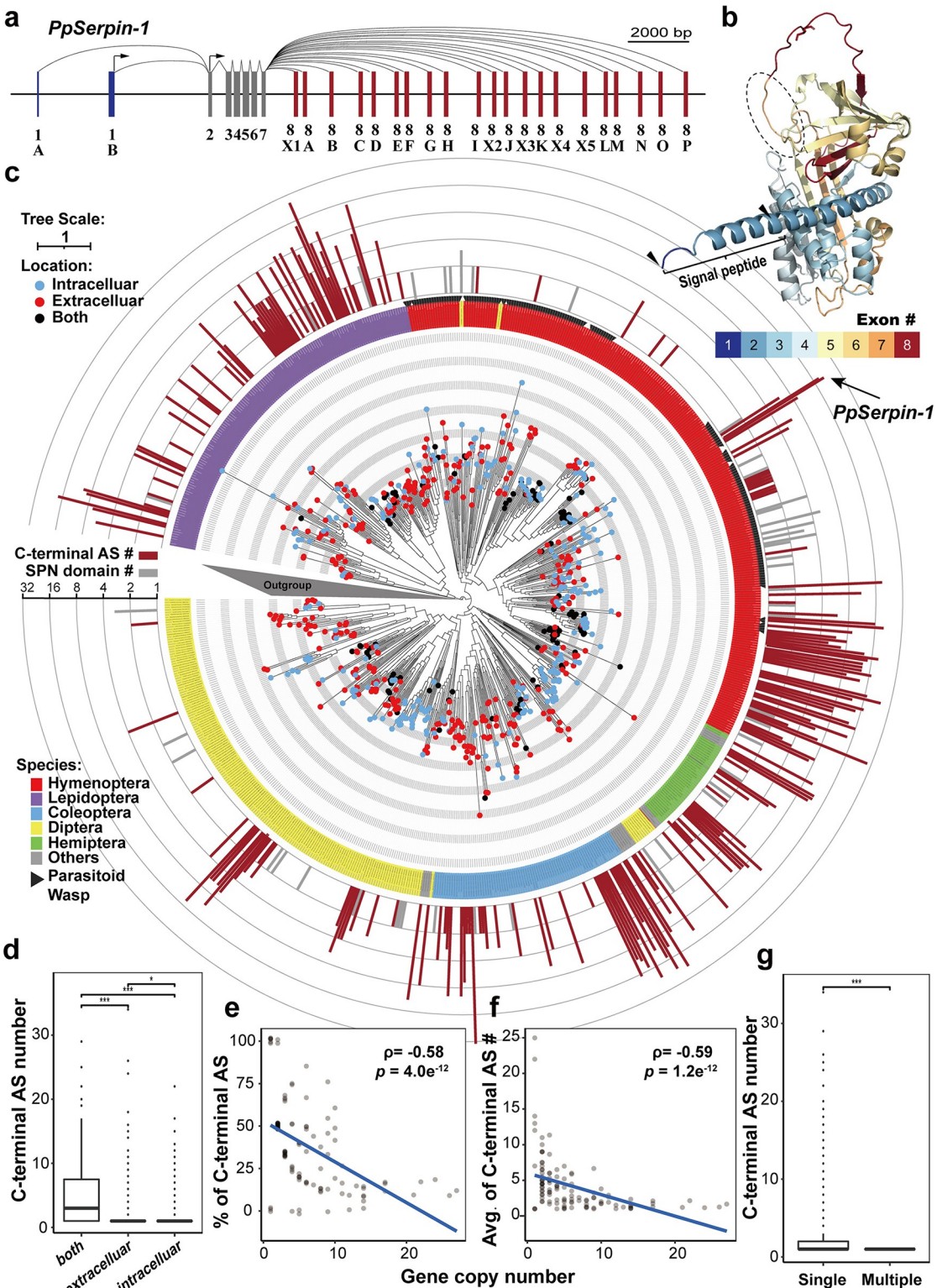

**Fig 1. Alternative splicing (AS), gene duplication (GD) and domain duplication contribute to serpin protein diversity in insects. (a)** Gene structure of *PpSerpin-1*. Gray indicates constitutive exons. Blue and red indicate N-terminal and C-terminal alternative exons, respectively. **(b)** Protein structure of PpSerpin-1F predicted by AlphaFold2. PpSerpin-1F is the longest protein isoform of the *PpSerpin-1* gene and was chosen as the representative. The gradient colors from blue to red indicate different exons. The bracket indicates the signal peptide. Black arrows indicate alternative translation start sites. The dashed circle indicates the

hinge region of PpSerpin-1F. **(c)** Phylogenetic tree of serpin in insects. The different colors on the labels indicate different insect orders. Black arrows indicate parasitoid wasps. The red and blue dots on branch terminals indicate the presence (extracellular) and absence (intracellular) of the signal peptide, respectively. Black dots on branch terminals indicate N-terminal AS with both extracellular and intracellular isoforms. The red bars indicate the log2 transformed numbers of C-terminal AS events. The gray bars indicate the log2 transformed numbers of serpin domains within a gene. **(d)** Relationship between the number of C-terminal AS events and gene localization. **(e)** Relationship between the percentage of C-terminal alternative spliced genes and serpin gene copy number. **(f)** Relationship between the number of C-terminal AS events and serpin gene copy number. **(g)** Relationship between the number of C-terminal AS events and sperin domain number. *** $p < 0.001$; * $p < 0.05$.

the N-terminus, one form was predicted to be secreted extracellularly with a signal peptide, while the other was predicted to localize intracellularly without a signal peptide (Fig 1B). At the C-terminus, AS occurs at the end of the serpin hinge region with an extra nucleotide G (Figs 1B and S1).

To construct the phylogeny of the *PpSerpin-1* gene, a homology search was performed against the insect Refseq_protein database in NCBI using the constitutive region of *PpSerpin-1* (for details, see Materials and Methods), resulting in a total of 1,230 matched genes. Phylogenetic analyses showed that the C-terminal AS of serpin genes was clustered into one clade with 731 genes but was relatively rare in the outgroup (Fig 1C). The following analyses were all focused on this clade with enriched C-terminal AS.

Both N-terminal and C-terminal AS are widespread in insects (Fig 1C). Out of 731 genes in this clade, 261 genes (35.7%, not including *PpSerpin-1*) have either N-terminal or C-terminal AS. At the N-terminus, 153 out of 731 (21.0%) had N-terminal AS, with 115 out of 153 having both extracellular and intracellular forms. These genes with both extracellular and intracellular forms at the N-terminus had more C-terminal AS than genes with only one N-terminal form, either extracellular or intracellular (Fig 1D; Mann–Whitney U test (MWUT); both comparisons: $p < 0.001$). At the C-terminus, 192 out of 731 (26.3%) genes had C-terminal AS, producing 1406 serpin proteins, with an average of 7.32 serpin proteins per gene. Notably, the number of C-terminal AS events showed rapid fluctuation, suggesting a high turnover rate of exon gain and loss (Fig 1C).

Together, AS, GD and domain duplication within genes contribute to serpin protein diversity. Forty-five genes have multiple serpin domains, producing 105 serpin domains with an average of 2.33 serpin domains per gene. The remaining 494 genes, which have neither C-terminal AS nor multiple serpin domains, likely originated by GD. Taken together, AS, GD and domain duplication accounted for 70.1%, 24.6% and 5.2% of the total number of 2005 serpin proteins/domains, respectively. Moreover, AS is negatively correlated with GD and domain duplication. The number of duplicated serpin gene copies in a species is inversely correlated with the percentage of C-terminal AS (Fig 1E; Spearman correlation; $\rho = -0.58$, $p = 4.0e^{-12}$) and with the means of C-terminal variants (Fig 1F; Spearman correlation; $\rho = -0.59$, $p = 1.2e^{-12}$). Both correlations held after phylogenetic correction (both tests: $p < 0.001$). In addition, AS and domain duplication within genes are mutually exclusive, as no multiple-serpin gene contains C-terminal AS (Fig 1G; MWUT; single- vs multiple-domain serpin: $W = 19755$, $p = 4.8e^{-5}$).

## AS genes show divergent sequence features from non-AS genes

Both AS and GD are major sources of protein diversity. To compare their differences in gene sequence characteristics, we divided the single-domain serpins into genes with C-terminal AS (referred to as the AS gene set) and those without (referred to as the non-AS gene set). Here, we focused on C-terminal but not N-terminal AS, as N-terminal AS often changes protein localization but not the mature serpin protein sequence.

For the AS gene set, the splicing positions of C-terminal AS occurred mainly near the end of the hinge region (GSEAAAVT in PpSerpin-1) (Fig 2A). Considering the last nucleotide at the end of the hinge region as position 0, 161 out of 192 (83.9%) AS genes spliced at position +1 and 19 (9.9%) at position +4. The AS gene set was more likely to splice at position +1 than the non-AS gene set (Fig 2A; $\chi^2 = 35.81$, $p = 0.00001$). Both AS and non-AS gene sets have G| GTAAGT and TTNCAG|N sequence motifs near C-terminal splicing sites (Fig 2B). Position 0 (at codon position 3), +5 (at codon position 2) and +11 (at codon position 2) are more inclined to be G, T and T in the AS gene set than in the non-AS gene set, respectively (Fig 2B; $p < 0.05$).

For the protein sequence, we divided the serpin proteins into constitutive (present in all isoforms) and C-terminal alternative spliced regions based on the end of the hinge region (Fig 2B). Conservation scores were estimated based on alignments and then mapped to PpSerpin-1F as the reference (Fig 2C). In the constitutive region, the AS gene set was more conserved than the non-AS gene set (Fig 2C and 2D; Wilcoxon signed-rank test (WSRT); $V = 985$, $p < 2.2e^{-16}$). Conversely, in the alternative region, the AS gene set was more variable than the non-AS gene set (Fig 2C and 2D; WSRT: $V = 1125$, $p = 1.5e^{-05}$). To exclude the effect of reference sequence selection, we also used non-AS genes within the clade (Dmel_SPN55B, NP_524953.1) or in the outgroup (Dmel_SPN27A, NP_001260143.1) as reference sequences, and the conclusions held regardless of reference selection (all comparisons: $p < 0.001$).

Furthermore, we compared the regulatory sequences of the AS and non-AS gene sets. For RNA-binding motifs, no motifs were enriched in the AS gene set compared to the non-AS gene set ($p > 0.05$), which may be due to the general short length of RNA-binding motifs and lack of statistical potency, as previously reported [44]. For binding motifs of transcription factors, two motifs were enriched in the AS gene set compared to non-AS. They were M02712_2.00 dl (dorsal) (Fig 2E; $p = 0.010$, enrichment ratio 8.01) and M03953_2.00 Dif (Dorsal-related immunity factor) ($p = 0.015$, enrichment ratio 7.39). dl and Dif are both transcription factors involved in the Toll pathway [45]. Compared to shuffled random sequences, dl and Dif binding motifs are significantly enriched in the AS gene set (dl: $p = 3.33e^{-7}$; Dif: $p = 7.70e^{-6}$) but not the non-AS gene set ($p > 0.05$). In addition, the probabilities of the presence of dl and Dif binding motifs are significantly correlated with the numbers of C-terminal AS (Fig 2E; Spearman correlation; dl: $\rho = 0.48$, $p < 2.2e^{-16}$, dif: $\rho = 0.48$, $p < 2.2e^{-16}$). These correlations are significantly higher than correlations using shuffled sequences, which have the same lengths as the true gene sequences (both comparisons: $p < 0.0001$). These results suggest that genes with more C-terminal AS are more likely to contain dl and Dif binding motifs, and this is not simply due to their longer sequence lengths.

We further asked how these dl and Dif binding motifs are distributed in sequences. We defined 6 region categories according to their strands (on coding or noncoding strands) and location (in promoter, exon or intron regions). Most of the best-hits were in the intron region of the coding strand for the AS gene set, while most of the best-hits were in the exon region of the noncoding strand for the non-AS gene set (Fig 2F). Best-hits of dl and Dif binding motifs are more likely to locate in the intron region of the coding strand of the AS gene set than that of the non-AS gene set (Fig 2F; dl: $\chi^2 = 50.40$, $p < 0.00001$; dif: $\chi^2 = 55.38$, $p < 0.00001$) and shuffled control sequences of the AS gene set (dl: $\chi^2 = 10.8984$, $p = 0.000962$; dif: $\chi^2 = 7.9206$, $p = 0.004888$). In addition, when we restricted the motif search to one of the six region categories, the correlations between the probabilities of the presence of dl and Dif binding motifs and the numbers of C-terminal AS were significantly higher than correlations using shuffled sequences in the intron region of the coding strand but not in the other five region categories (dl: $z = 2.7$, $p = 0.0065$; dif: $z = 4.4$, $p < 0.0001$; all other comparisons: $p > 0.05$). Therefore, we conclude that dl and Dif binding motifs tend to be enriched in the intron region of the coding strand within the AS gene set.

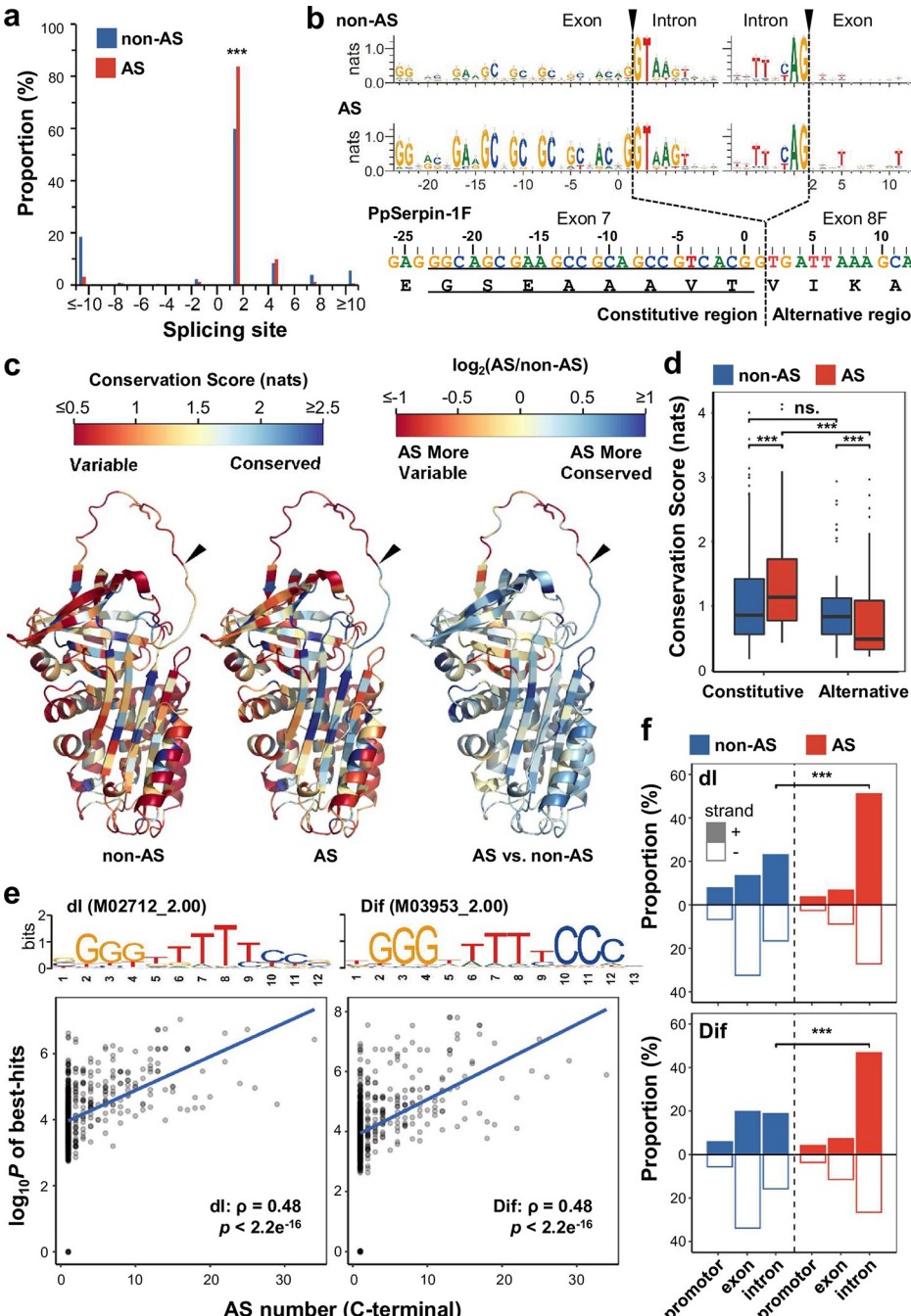

**Fig 2. AS genes show divergent sequence features from non-AS genes.** (**a**) Distribution of splicing sites on the AS and non-AS gene sets. The AS gene set represents single-domain serpin genes with C-terminal AS, and the non-AS gene set represents single-domain serpin genes without C-terminal AS. The end of the hinge region is noted as position 0. (**b**) Sequence logos of AS and non-AS gene sets. The dashed lines indicate splicing positions. The underline indicates the hinge region. (**c**) Predicted protein structure of PpSerpin-1F with mapped conservation scores of AS and non-AS gene sets. (**d**) Comparisons of conservation scores between AS and non-AS genes. (**e**) Correlation of C-terminal AS number and probability of dl and Dif binding motif best hits. (**f**) Binding motif hit distribution of dl and Dif on AS and non-AS genes. + and—indicate coding and noncoding strands, respectively. *** $p < 0.001$; ns.: $p > 0.05$.

## Parasitoid wasps employ fewer AS in serpin, but *PpSerpin-1* is an expansion of the AS exon number

We further investigated the potential relationship between AS and the parasitic life strategy. Higher numbers of total serpin proteins/domains per species were found in parasitoid wasps than in nonparasitoid hymenopterans (Fig 3A; $W = 370$, $p = 0.0026$). Compared to nonparasitoid hymenopterans, more GD (Fig 3B; MWUT; $W = 434$, $p = 5.5e^{-6}$) and domain duplication (S2 Fig; MWUT; average: $W = 362.5$, $p = 2.1e^{-4}$; proportion: W = 350, p = 8.1e$^{-4}$) but less AS (Fig 3C and 3D; MWUT; average: $W = 96$, $p = 0.0022$; proportion: $W = 96$, $p = 0.0017$) are utilized in parasitoid wasps to produce serpin protein/domain diversity. Consistent with this, gene expansions in parasitoid wasps are generally more recent (S3A Fig; MWUT; $W = 11429$, $p = 3.0e^{-6}$), with a higher proportion of genus-specific and family-specific expansions (Fig 3E; genus-specific: $\chi^2 = 14.09$, $p = 1.74e^{-4}$; family-specific: $\chi^2 = 10.49$, $p = 1.20e^{-3}$) than those in nonparasitoid hymenopterans. Conversely, exon expansions in parasitoids are generally more ancient (S3B Fig; MWUT; $W = 16338$, $p = 1.25e^{-4}$), with a lower proportion of lineage-specific exon expansions (Fig 3F; $\chi^2 = 20.14$, $p < 0.00001$) than those in nonparasitoid hymenopterans. Patterns are similar when comparing parasitoid wasps with all nonparasitoid insects (not limited to Hymenoptera) (all comparisons: $p < 0.05$).

However, *PpSerpin-1* shows an expansion of AS number compared to other parasitoids. The C-terminal AS numbers of the *PpSerpin-1* and *Nasonia vitripennis* LOC100122505 genes are increased compared to their homologous genes in the Chalcidoidea wasps, i.e., *Trichogramma pretiosum*, *Ceratosolen solmsi marchali*, and *Copidosoma floridanum* (Fig 3G). For *PpSerpin-1* alternative exons, H and X2 clustered together with the XP_008201831.1-specific exon of the *N. vitripennis* LOC100122505 gene as the closest outgroup, suggesting *PpSerpin-1*-specific exon expansion after divergence from the common ancestor with the *N. vitripennis* LOC100122505 gene, which is estimated to have occurred approximately 19 million years ago (MYA) [46]. Sixteen out of 21 alternative exons show clear orthologous relationships with alternative exons of the *N. vitripennis* LOC100122505 gene, suggesting that most alternative exons of *PpSerpin-1* existed prior to the divergence between *Pteromalus* and *Nasonia*.

We then estimated the pairwise substitution rates of orthologous exons between the *PpSerpin-1* and *N. vitripennis* LOC100122505 genes. Compared to constitutive exons, alternative exons show higher nonsynonymous substitution rates (*dN*) (S4B Fig; MWUT; $W = 15$; $p = 0.013$) and lower synonymous substitution rates (*dS*) (S4C Fig; MWUT; $W = 84$; $p = 0.0061$), resulting in higher *dN*/*dS* values (S4A Fig; MWUT; $W = 5$; $p = 5.1e^{-4}$). These results suggest positive selection diversifying protein sequences in alternative exons with lower substitution rates in synonymous sites, which can be important in the regulation of the AS process [1,2,47].

## *PpSerpin-1* is involved in the wasp's immune response and recruited into both wasp venom and larval saliva

Next, we investigated the function of isoforms of *PpSerpin-1*. First, isoform-specific RT–PCR confirmed the presence of all 21 C-terminal AS forms of *PpSerpin-1* (Fig 4A). These isoforms were more likely to be upregulated by the gram-positive bacterium *Micrococcus luteus* and the fungus *Beauveria bassiana* than by PBS or the gram-negative bacterium *Escherichia coli* (Fig 4B; WSRT; all comparisons: $p < 0.001$).

In our previous study, PpSerpin-1O was isolated from *P. puparum* venom [22]. RNA-seq data showed that multiple *PpSerpin-1* isoforms are expressed in the venom gland (Fig 4C). We also re-analyze the published *P. puparum* venom proteomic data [37] and identified isoforms B, G, O and P. By Western blotting, PpSerpin-1 and its homologous proteins were detected in

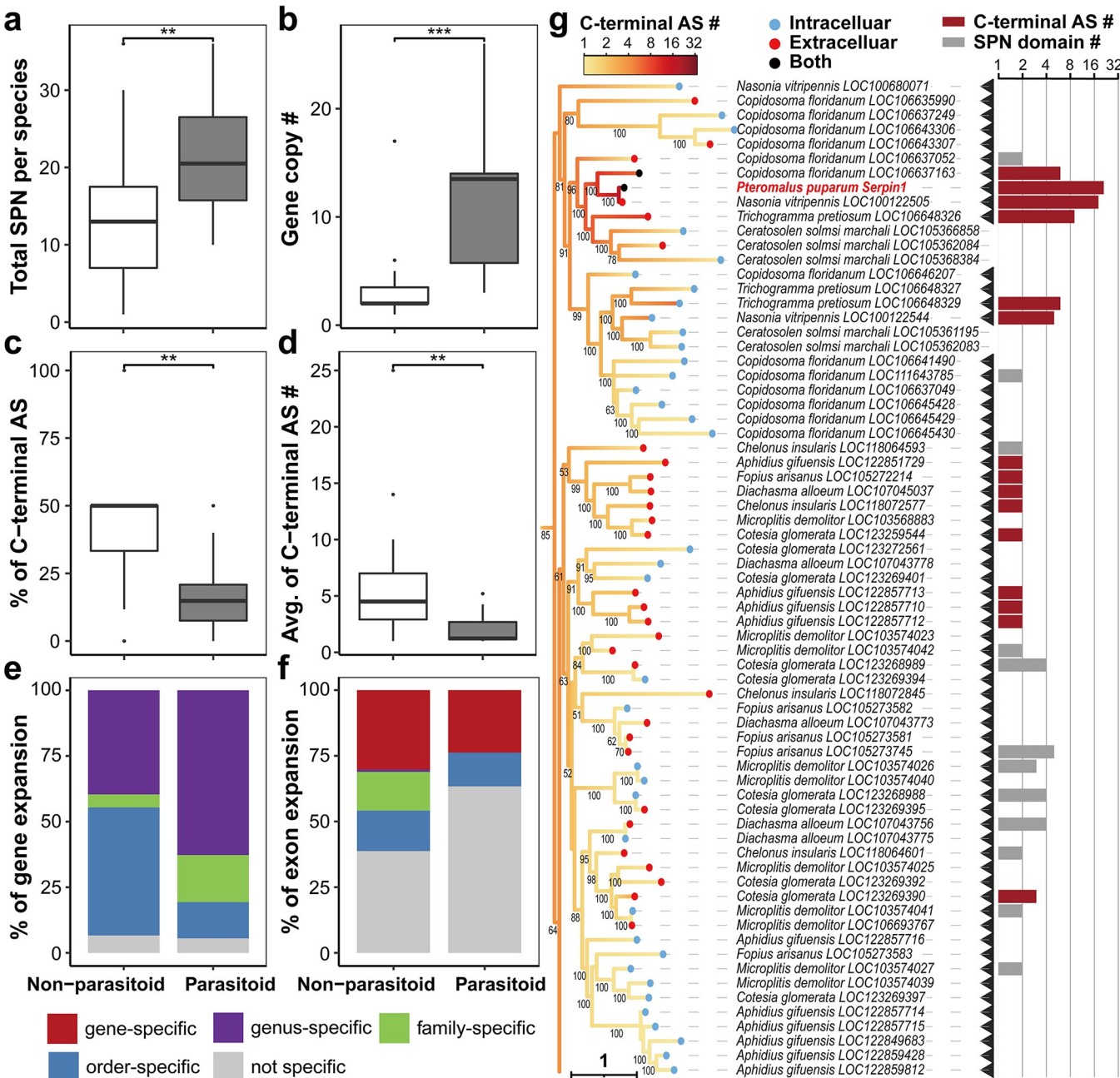

**Fig 3. Parasitoid wasps employ fewer AS events in serpin, but *PpSerpin-1* shows an expansion of the C-terminal AS number. (a-f)** Comparison between parasitoid wasps and nonparasitoid hymenopterans of (a) total serpin protein per species, (b) gene copy number, (c) percentage of genes with C-terminal AS, (d) average C-terminal AS number, (e) distribution of gene expansion and (f) distribution of exon expansion. **(g)** Expansion of the C-terminal AS number in *PpSerpin-1*. Black arrows indicate parasitoid wasps. The red and blue dots on branch terminals indicate the presence (extracellular) and absence (intracellular) of the signal peptide, respectively. Black dots on branch terminals indicate N-terminal AS with both extracellular and intracellular isoforms. The gradient color on branches indicates the estimated ancestral states of C-terminal AS numbers. The red bars indicate the log2 transformed numbers of C-terminal AS events. The gray bars indicate the log2 transformed numbers of serpin domains within a gene. *** $p < 0.001$, ** $p < 0.01$.

the venoms of *P. puparum* and its relative species *N. vitripennis*, *Muscidifurax raptor*, and *M. uniraptor* (Fig 4D). Additionally, by examining published venom sequences, we also identified PpSerpin-1 homologous proteins in the venom of *Trichomalopsis sarcophagae* and *Urolepis*

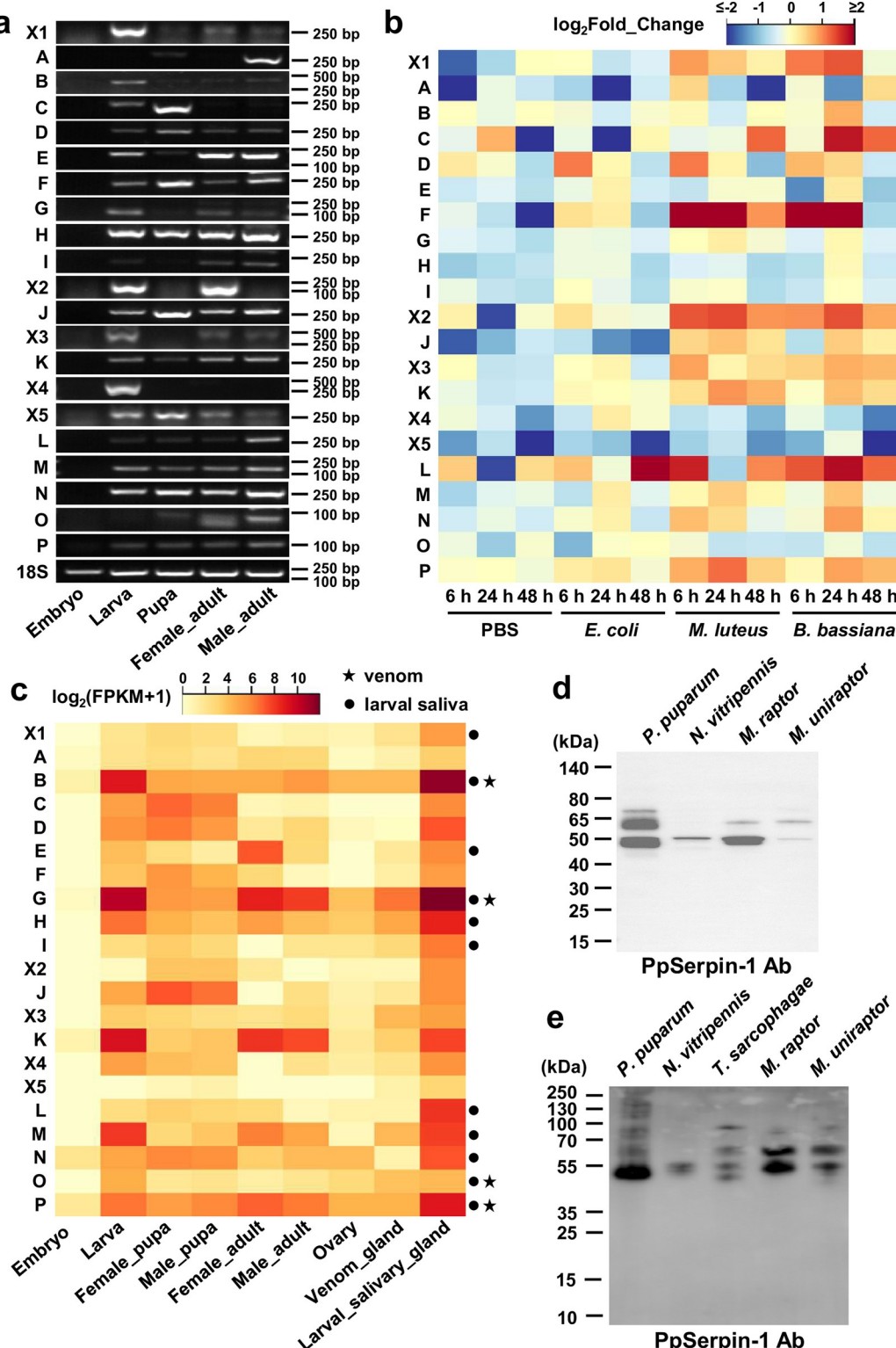

**Fig 4. *PpSerpin-1* isoforms are involved in the wasp's immune response and parasitism.** (**a**) Isoform-specific RT–PCR of *PpSerpin-1* isoforms. (**b**) Expression of *PpSerpin-1* isoforms in response to injection of PBS, *E. coli*, *M. luteus* or *B. bassiana*. Expression levels were calculated from RNA-seq data. (**c**) Expression of *PpSerpin-1* isoforms in different developmental stages and tissues. Black dots indicate proteomic identification in *P. puparum* larval saliva. Black stars indicate proteomic identification in venom of *P. puparum* female wasps. (**d**) Western blot detection in venom of *P.*

*puparum*, *Nasonia vitripennis*, *Muscidifurax raptor* and *M. uniraptor* using PpSerpin-1 antibodies. (**e**) Western blot detection in larval saliva of *P. puparum*, *Nasonia vitripennis*, *Trichomalopsis sarcophagae*, *Muscidifurax raptor* and *M. uniraptor* using PpSerpin-1 antibodies.

*rufipes*, exhibiting 90.1% and 91.6% identity to the constitutive sequence of PpSerpin-1 protein, respectively [17,35].

In addition, both RT–PCR and transcriptomic results showed that some *PpSerpin-1* isoforms were highly expressed in the larval stage (Fig 4A and 4C), particularly in the larval salivary gland (Fig 4C). We therefore hypothesize that some isoforms have been recruited into *P. pupuarum* larval saliva, which has been demonstrated to suppress host melanization [48]. Consistent with this hypothesis, PpSerpin-1 proteins were detected by Western blot using PpSerpin-1 antibodies in PBS incubated with *P. puparum* larvae (S5 Fig) and in larval saliva of *P. puparum* (Fig 4E). Protein homologs of PpSerpin-1 were also detected in the larval saliva from relatives of *P. puparum*, i.e., *N. vitripennis*, *T. sarcophagae*, *Muscidifurax raptor* and *M. uniraptor* (Fig 4E). Moreover, we identified 11 isoforms of *PpSerpin-1*, i.e., X1, B, E, G, H, I, L, M, N, O and P, in the larval saliva of *P. puparum* by the protomic approach (Fig 4C).

## *PpSerpin-1* isoforms inhibit host melanization immunity

To test the function of *PpSerpin-1* in larval saliva of *P. puparum*, we demonstrated that *P. puparum* larval saliva inhibited the host's hemolymph melanization (Fig 5A; Dunnett's test; $p = 0.0038$), and this inhibitory effect could be eliminated by the antibody of PpSerpin-1 (Fig 5A; Dunnett's test; $p > 0.05$). To further investigate the function of *PpSerpin-1* in suppressing host melanization, recombinant proteins of each isoform were produced (S6 Fig). PpSerpin-1 isoforms A, G, O and P significantly inhibited hemolymph melanization of *P. rapae* (Fig 5B; Dunnett's test; all comparisons: $p < 0.001$) in a dose-dependent manner (S7 Fig) and formed complexes with host hemolymph proteins in pull-down assays (Fig 5C). Pull-down assays were also performed for the remaining isoforms, which were identified in wasp venom or larval saliva. PpSerpin-1B and H formed complexes with host hemolymph proteins, although they showed no inhibition of host melanization (Fig 5C).

To identify the targets of isoforms A, B, G, H, O, and P, we cut gel slices covering the complexes ranging from ~60 to 100 kDa and identified 59, 9, 56, 29, 42, and 86 host proteins, respectively (S3 Table). Among them, six identified proteins were serine proteases, i.e., PrPAP1, PrPAP3, PrHP8, PrHP17, PrHP and PrHP1 (Fig 5D). The nomenclature of *Pieris rapae* hemolymph proteases (PrHPs) was based on their orthology with *Manduca sexta* hemolymph proteases [49] (S8 Fig).

Among these protease candidates, PrPAP1 is *P. rapae* prophenoloxidase-activating proteinase 1 and is critical for hemolymph melanization. The presence of PrPAP1 in pull-down samples of PpSerpin-1A, O and P was confirmed by Western blotting using PrPAP1 antibodies (Fig 6A). In vitro inhibitory assays showed that isoforms A, O and P but not G significantly inhibited the activity of recombinant PrPAP1 (Fig 6B; Dunnett's test, for A, O and P: $p < 0.001$; for G: $p = 0.94$). Consistent with this, isoforms A, O, and P but not G formed SDS–PAGE stable complexes with recombinant PrPAP1 (Figs 6C, 6D and S9). Presumably, two identified peptides of PrPAP1 in the pull-down sample of PpSerpin-1G may be leaks from the neighboring pull-down samples of PpSerpin1-A and O. The stoichiometries of inhibition (SIs) of PrPAP1 by PpSerpin1-A and P were 13.37 and 197.33, respectively (Fig 6E and 6F), which were larger than the previously reported SI of 2.3 by PpSerpin-1O [22]. These results demonstrate that PpSerpin-1 isoforms A, O, and P suppress host melanization immunity by inhibiting PrPAP1.

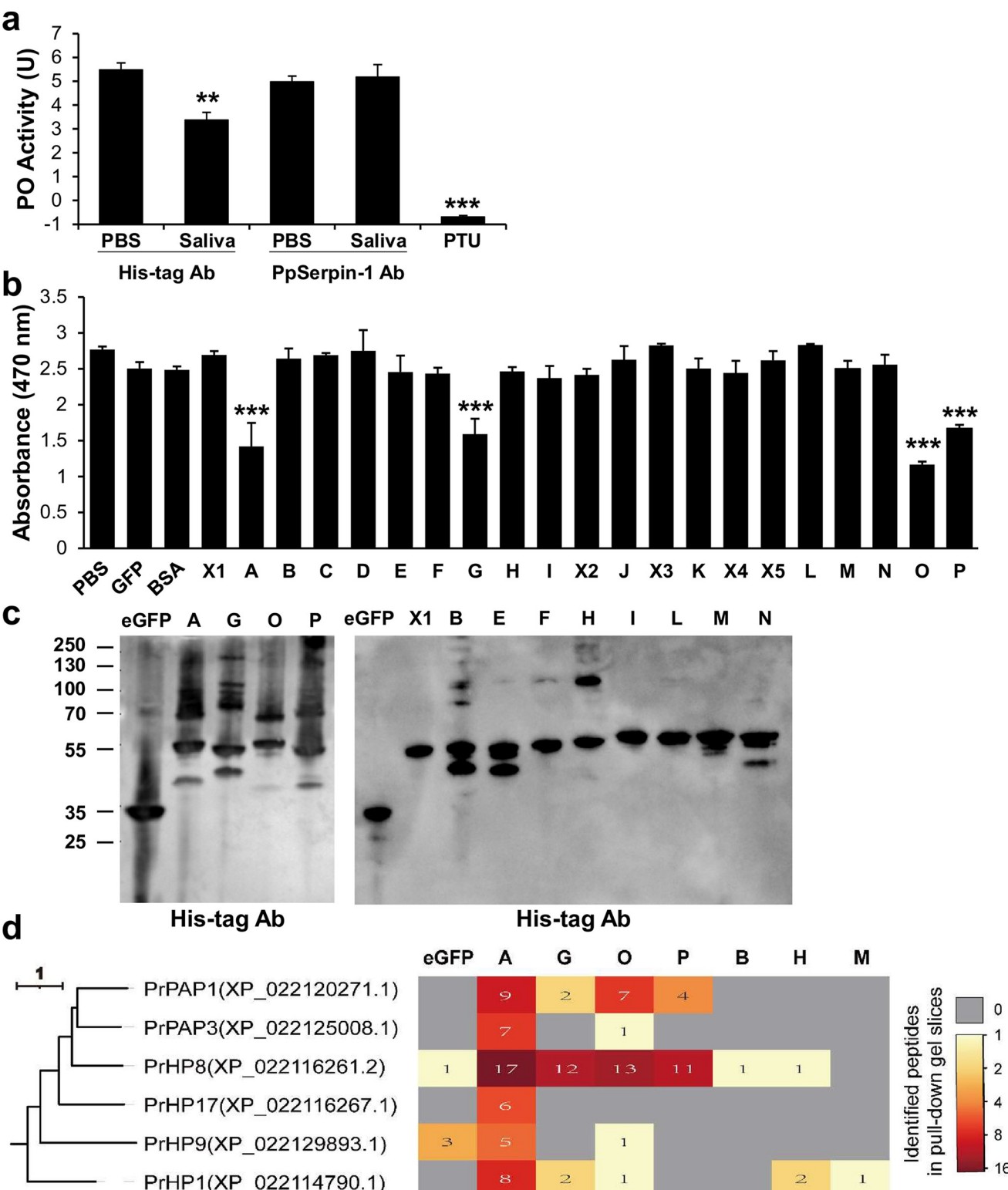

**Fig 5. PpSerpin-1 isoforms inhibit host melanization and form complexes with host hemolymph proteins. (a)** Effect of *P. puparum* larval saliva and PpSerpin-1 antibody on PO activity of host *P. rapae* hemolymph. **(b)** Effect of PpSerpin-1 isoforms on PO activity of host *P. rapae* hemolymph. **(c)** Pull-down assays against host hemolymph using PpSerpin-1A, G, O, P and other isoform proteins identified in venom or larval saliva of *P. puparum*. **(d)** Host hemolymph proteinases identified in pull-down samples of PpSerpin-1 isoforms. *Pieris rapae* hemolymph proteinases (PrHPs) are named based on their orthology with *Manduca sexta* hemolymph proteinases [49]. Pull-down samples of eGFP and PpSerpin-1M were used as negative controls. *** $p < 0.001$, ** $p < 0.01$.

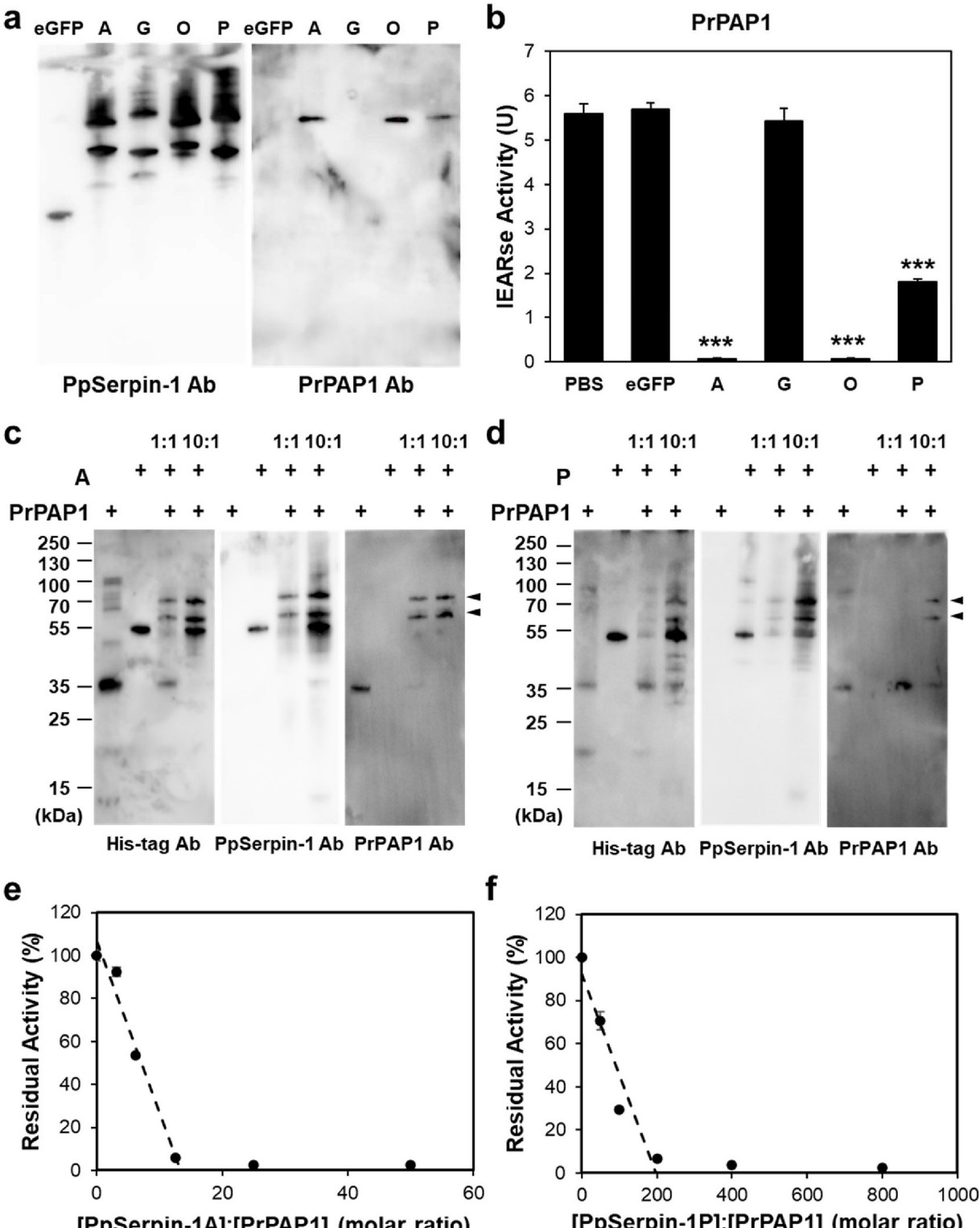

**Fig 6. PpSerpin-1 isoforms A, O, and P but not G inhibit host PrPAP1.** (a) Western blot detection of PrPAP1 in pull-down samples of PpSerpin-1A, G, O and P. (b) Effect of PpSerpin-1A, G, O and P on PrPAP1 activity. *** $p < 0.001$. (c) SDS-stable complex formation between PpSerpin-1A and PrPAP1. (d) SDS-stable complex formation between PpSerpin-1P and PrPAP1. (e) Stoichiometry of PrPAP1 inhibition by PpSerpin-1A. (f) Stoichiometry of PrPAP1 inhibition by PpSerpin-1P.

## Discussion

In this study, we report the divergent features of AS and GD in the evolution of insect serpins and their potential associations with the parasitic life strategy. We also found that *PpSerpin-1* shows a number expansion of AS exons, which is involved in the wasp's immune response and recruited to both wasp venom and larval saliva to suppress host immunity.

Not all isoforms recruited into wasp venom or larval saliva inhibit host melanization. These isoforms may serve other functions, such as regulating the host's Toll pathway or modulating host metabolism. Proteomic identification results from the pull-down assays of these isoforms can provide hints into their targets and functions, which will be an interesting future direction. Moreover, several isoforms identified in larval saliva failed to form complexes with host hemolymph proteins. These *PpSerpin-1* isoform proteins may target proteins out of the host hemolymph, e.g., in host hemocytes or gonad cells, inside host guts, or regulate proteases from wasp larval saliva. Alternatively, the inconsistency between proteomic identification and functions of *PpSerpin-1* isoforms may be simply due to leaky expression from the noise of AS regulation [50]. More investigations are needed to test these hypotheses.

In addition to AS, GD and domain duplication jointly contribute to the protein diversity of insect serpins. Among them, AS and GD are the major sources of protein diversity. We observed an inverse relationship between the production of AS variants and gene family size across different species within a serpin clade. This is similar to previous reports that the production of AS variants was inversely correlated with the size of the gene family within the same species [9,10,51]. These results suggest negative regulatory mechanisms between AS and GD in the production of protein diversity.

AS is an efficient mechanism to generate significant protein diversity without requiring duplication and divergence of the entire gene [2,3,8]. In serpin AS genes, C-terminal AS tends to occur at the end of the hinge region with an extra nucleotide G. The serpin hinge region is critical to the conformational change and thus the inhibitory function of serpin [30,31,33,34], while the extra nucleotide G may be important in the splicing process of AS. The splicing sites in AS serpins reflect the maximum reuse of sequence. Compared to non-AS genes, AS genes are more conserved in the constitutive region but more variable in the alternative region. In addition, in the comparison of *PpSerpin-1* with its orthologous gene LOC100122505 in *N. vitripennis*, the alternative exons show significantly higher nonsynonymous substitution rates than the constitutive exons. Collectively, these results imply that AS allows for distinct regions to experience divergent evolutionary pressures, enabling rapid protein evolution in serpin RCL while preserving most of the functional backbone under strong conservation.

On the other hand, the regulation of AS is often complicated [52]. For pairwise substitution rates between *PpSerpin-1* and its orthologous gene LOC100122505 in *N. vitripennis*, alternative exons show significantly lower synonymous substitution rates than constitutive exons, suggesting that synonymous sites may be critical in the regulation of AS and thus under higher selective pressure in alternative exons [1,2,47]. In addition, the binding motifs of two transcription factors, dl and Dif, are enriched in the AS gene set compared to the non-AS gene set, particularly preferring the intron region of the coding strand of AS genes. Genes with more C-terminal AS are more likely to contain dl and Dif binding motifs. Accumulated evidence suggests that transcription factors can be directly or indirectly involved in the regulation of AS, although the mechanisms are not fully understood [53,54]. As pre-mRNA splicing mainly takes place during transcription, extensive crosstalk has been reported between these two processes [53]. One possible explanation could be that some expanded serpin isoforms are involved in the Toll pathway. The expression of these isoforms might be regulated by Toll-pathway-related dl and Dif, probably through binding to the intron regions of serpin gene pre-mRNA.

To manipulate their hosts, parasitoid wasps often recruit effector genes, e.g., venom or larval salivary genes [14,15]. Serpin, a common component of parasitoid venom [17,20–23,35–43], has also been reported in teratocytes of parasitoid wasps for host regulation [55,56]. Here, we further extend serpin recruitment to wasp larval saliva. A previous study also reported that *P. puparum* larval saliva inhibits host immunity [48]. In light of this, the higher numbers of serpin protein/domains in parasitoid wasps may be related to the recruitment of host effector genes. Another example of an AS venom gene is LOC122512947 from *Leptopilina heterotom* with 2 N-terminal and 20 C-terminal AS isoforms [36]. In addition, *Leptopilina boulardi* LbSPNy (ACQ83466.1) was reported as a venom non-AS gene that inhibits host melanization [43]. LbSPNy forms a *Leptopilina*-specific non-AS gene cluster with LOC122509667, 122505241, 12200434, 122502279, 122503460, and 122502269 of *L. heterotoma*, suggesting that LbSPNy likely originated from GD. Extensive GD of the serpin gene was also reported in the venom of *M. mediator* [21]. Our findings further show that the increased number of serpins in parasitoids results from more GD but less AS. This is consistent with the general observation that venom genes in parasitoid wasps and other venomous animals are more likely to originate from GD than AS [14,57].

In *PpSerpin-1*, we observed a number expansion of alternative exons compared to other parasitoid serpin genes. Reconstructing the accurate evolutionary history of these exons is challenging, particularly for ancient expanded exons, due to their accelerated protein evolution and short sequences (44~50 amino acids for PpSerpin-1 alternative regions). Phylogenetic tree analysis revealed that alternative exons of PpSerpin-1H and X2 form a gene-specific cluster, suggesting a recent *Pteromalus*-specific exon duplication. However, the majority (16 out of 21) of *PpSerpin-1* alternative exons show clear orthologous relationships with those of the *N. vitripennis* LOC100122505 gene. One possible explanation is that these exons underwent lineage-specific exon duplications before the divergence of *Pteromalus* and *Nasonia* (~ 19 MYA) [46] and were retained in both species, probably serving conserved and essential functions. The expanded alternative exons in *PpSerpin-1* may contribute to its ecological adaptation. Consistent with this hypothesis, we found that several isoforms of *PpSerpin-1* are involved in the wasp's immune response, have been recruited to both wasp venom and larval saliva, and suppress host immunity.

However, no obvious expressional specialization, especially to venom glands, was observed for *PpSerpin-1* isoforms. We speculate that accurate expressional regulation is an obstacle to the recruitment of host effector genes by AS, particularly that *PpSerpin-1* is likely involved in the regulation of wasp self-immunity. Several *PpSerpin-1* isoforms can be upregulated by *M. luteus* and *B. bassiana*, suggesting that *PpSerpin-1* may be involved in the wasp's Toll pathway, which is preferentially activated by gram-positive bacteria and fungi in *Drosophila* [45]. In AS genes, differential expressional regulation between self and host manipulation functions may be difficult to evolve. On the other hand, expressional divergence of duplicated genes often occurs after location segregation of gene copies [58], presumably due to changes in cis-regulatory regions. The positional linkage of alternative exons makes the expressional regulation of AS more challenging and requires more complicated regulatory mechanisms. In particular, high expressional specialization should be required to avoid self-harm for toxic virulent genes. This may explain why AS is less employed than GD in the recruitment of parasitoid host effector genes.

In summary, we show that both AS and GD contribute to the evolution of insect serpin with differential features. We report that a parasitoid serpin gene has evolved through exon number expansion of AS and show its involvement in wasp's immunity and recruitment into the wasp's venom and larval saliva to manipulate host immunity.

## Materials and methods

### Insect rearing

Laboratory cultures of *P. puparum*, *N. vitripennis*, *T. sarcophagae*, *M. raptor* and *M. uniraptor* were maintained in *Drosophila* tubes at 25°C with a photoperiod of 14:10 h (light:dark) as previously described [17,37]. Pupae of *Pieris rapae* were used as hosts for *P. puparum*, and pupae of *Sarcophaga bullata* were used as hosts for *N. vitripennis*, *T. sarcophagae*, *M. raptor* and *M. uniraptor*. Once emerged, *P. puparum* adult wasps were fed *ad libitum* with 20% (v/v) honey solution to lengthen their life span.

### Alternative isoform identification for the *PpSerpin-1* gene

Using accumulated transcriptomic data [46], sequencing reads were filtered by Trimmomatic v0.38 [59], mapped to the *P. puparum* genome by TopHat v2.0.12 [60], and assembled into transcripts using Cufflinks v2.2.1 [60]. For exhaustive identification of *PpSerpin-1* isoforms, we further manually identified isoforms using IGV browser v2.3.91 based on mapped reads [61]. The expression levels of *PpSerpin-1* isoforms were estimated using Cufflink v2.2.1 [60].

### Protein structure prediction

The protein structure of PpSerpin-1F was modeled using AlphaFold2 with default settings [62,63]. No structural template was used. All five AlphaFold2 models were tested. Model 4 was selected as the best prediction based on its highest pLDDT score of 88.7. Alignemnts for protein modeling were generated through MMSeqs2 [64] against the UniRef+Environmental database.

### Sequence fetch and feature analyses

BLASTP [65] was performed using the constitutive protein sequence of the *PpSerpin-1* gene against the Refseq_protein database on NCBI (accessed at Dec 2021). The organism was limited to "Insecta (taxid:50557)" with a maximum target sequence of 5000 and an e-value of 1e$^{-5}$. One species per genus was selected as a representative to reduce sampling bias. GenBank files were retrieved for each gene using EFetch v0.2.2 (https://dataguide.nlm.nih.gov/edirect/efetch.html). Isoform sequences and splicing positions were extracted from GenBank files for each gene using homemade python scripts. For each gene, the number of N-terminal AS events was approximately estimated by counting the number of different sequences in the first 20 aa. Similarly, the number of C-terminal AS events was approximately estimated by counting the number of different sequences in the last 50 aa. Signal peptides were predicted using SignalP v5.0 [66]. Serpin domains were annotated using the NCBI Batch Web CD-Search Tool (https://www.ncbi.nlm.nih.gov/Structure/bwrpsb/bwrpsb.cgi) against the "CDD 58235 PSSMs" database [67] (accessed at Jan, 2022).

### Phylogenetic analyses

For gene phylogeny, the longest protein isoform was selected for each gene. Proteins were aligned using Mafft v7.310 [68]. The gene tree was constructed using IQTree v2.2.0 with an ultra bootstrap of 1000 [69]. The best-fit model (LG+R10) was automatically selected by built-in ModelFinder [70] in IQTree. For exon phylogeny, annotated serpin domains were extracted with a C-terminal extending 30 aa or to the end of the sequences if less than 30 aa. For genes with C-terminal AS, only nonredundant C-terminal sequences were included. Serpin domain sequences were aligned using Mafft v7.310 [68]. The tree was constructed using IQTree v2.2.0 [69]. To reduce sampling bias, one species per genus was selected as a representative for

visualization and subsequent statistical analyses. Ancestral character estimation was performed for the C-terminal AS number of each internal node using the fastAnc function in the R package "phytools" v1.2.0 [71]. Trees were pruned by Newick Utilities v1.6 [72] and visualized on iTOL [73].

### Expansion level analyses

For every pair of genes from the same species in the gene phylogeny, two genes are defined as species-specific expansion if all genes in the clade of the common ancestor of these two genes belong to the same species. If all genes in the clade of the common ancestor of these two genes from the same species belong to the same family, two genes are defined as family-specific expansion, and so on. Similarly, for every two alternative C-terminal exons from the same gene in the exon phylogeny, two exons are defined as gene-specific expansion if all exons in the clade of the common ancestor of these two exons belong to the same gene, and so on. Expansion levels were determined by homemade R scripts.

### WebLogo

Consensus sequence logos were created using WebLogo v3.7.9 [74]. For protein alignments, conservation scores were estimated for each site using WebLogo v3.7.9 [74]. Conservation scores of the corresponding positions in the reference sequence were extracted for comparisons. For the constitutive region of AS genes, the longest protein isoforms of each gene were selected as representatives. For the C-terminal alternative region of AS genes, all nonredundant sequences (based on the last 50 aa at the C-terminus) were used. Conservation scores were mapped to the PpSerpin-1F structure using PyMOL v2.5.0 (https://pymol.org).

### *dN* and *dS* estimation

For each orthologous exon pair between *PpSerpin-1* and LOC100122505 (the ortholog of *PpSerpin-1* in *N. vitripennis*), protein sequences were aligned using Mafft v7.310 [68] and then reverse translated to codons using PAL2NAL v14 [75]. If a codon crossed the boundary of two exons, the entire codon was counted in the exon that contained more nucleotide bases of that codon. Pairwise *dN* and *dS* values were estimated using PAML v4.9j [76].

### Motif scanning and enrichment analysis

Motif enrichment analyses were performed using SEA (Simple Enrichment Analysis) in MEME v5.5.0 [77,78]. Gene sequences with C-terminal AS were set as primary sequences, and shuffled sequences or gene sequences without C-terminal AS were set as control sequences. *Drosophila melanogaster* "CIS-BP 2.00 single species DNA" or "CISBP-RNA single species RNA" was set as the motif database. Fisher's exact tests were performed if the primary and control sequences had the same average length (within 0.01%); otherwise, binomial tests were performed [78]. Sequences of pomotor, exon and intron regions were extracted from GenBank files. Promoter regions were approximated by using the upstream sequences of the translation start sites. Exon regions were approximated using coding sequences by masking other regions using "N". For binding motifs of dl (M02712_2.00) and Dif (M03953_2.00), hits were detected with a *p* value threshold of 1 using FIMO in MEME [77]. Shuffled sequences were created by fasta-shuffle-letters in MEME [77].

## Specific RT–PCR

*Ptermoalus puparum* embryos (<10 hr after parasitism), larvae (combined 2–3 day larvae after parasitism), yellow pupae (mixed with female and male pupae), adult females and males (combined 1–5 day adults after emergence) were collected and rinsed with PBS. Total RNA was extracted using TRIzol reagent (Invitrogen, USA) according to the manufacturer's protocol and then reverse transcribed using TransScript One-step gDNA Removal and cDNA Synthesis Super-Mix (TransGen, Beijing, China) with random primers. Isoform-specific primers were designed to span constitutive exon 7 and alternative exon 8 using PerlPrimer v1.1.21 [79] and are listed in S1 Table. PCR was performed using TransTaq HiFi DNA Polymerase (TransGen, Beijing, China) with 20–35 cycles of amplification. PCR products were analyzed by electrophoresis on a 1% (g/mL) agarose gel and confirmed by Sanger sequencing (Sangon Biotech, Shanghai, China).

## Microbial stimulus of *P. puparum*

Freshly cultured *M. luteus*, *E. coli*, and *B. bassiana* were harvested by centrifugation and subsequently washed with sterile PBS three times. These microbes were suspended in PBS at a density of $5 \times 10^7$ cells/mL. Two-day old *P. puparum* female adults were selected and anesthetized using carbon dioxide. A sterilized acupuncture needle was immersed in the suspension containing either bacteria or fungi and then inserted into the inter-segmental space of the parasitoid wasp's abdomen. Wasps that were pricked with sterile PBS were used as a control group. Female wasps were then collected at 6, 24, and 48 h after the microbial stimulus.

## Wasp larval saliva collection

Saliva of wasp larvae was collected as previously described with minor modifications [48]. Briefly, wasp larvae were collected by opening host pupae 3 days after parasitism. Larvae were rinsed with PBS and electrically stimulated using an acupuncture device (Hwato, China) with a frequency of 50 Hz and a current of 2.5 mA. Secreted saliva drops at larval mothparts were transferred into PBS using pipette tips. The protein concentration was determined using a Pierce BCA Protein Assay Kit (Thermo Scientific, USA).

## LC–MS/MS

Fifty micrograms of protein from wasp larval saliva was digested by trypsin using the filter-aided sample preparation (FASP) method. After desalting by a C18 cartridge, the digestion product was lyophilized and redissolved in 40 μl of 0.1% formic acid solution. LC–MS/MS was conducted on an Easy nLC HPLC system (Thermo Scientific, USA) with a flow rate of 300 nl/min followed by Q-Exactive (Thermo Finnigan, USA). The sample was loaded on a Thermo Scientific EASY column (5 μm, 2 cm × 100 μm, C18) and then separated on another Thermo Scientific EASY column (3 μm, 75 μm × 100 mm, C18). Buffer A was water with 0.1% formic acid, and buffer B was 84% acetonitrile with 0.1% formic acid. The columns were first equilibrated with 95% buffer A, then from 0% to 35% buffer B in 50 min, from 35% to 100% buffer B in 5 min, and finally 100% buffer B for 5 min. The charge-to-mass ratios of peptides and fractions of peptides were collected 20 times after every full scan. The resulting MS/MS spectra were searched against the *P. puparum* genome database using Mascot 2.2 in Proteome Discoverver. "Carbamidomethyl (C)" was set as a fixed modification. "Oxidation (M)" and "Acetyl (Protein N-term)" were set as variable modifications. The maximum number of missed cleavages was set as 2. FDR $\leq$ 0.01 was set to filter the protein identification. The same software and parameters were used for the reanalysis of *P. puparum* venom proteomic data [37]. This part was conducted by Shanghai Applied Protein Technology Co., Ltd. (Shanghai, China).

## Western blot

Protein was separated by electrophoresis on 12% SDS–PAGE and transferred to a PVDF (polyvinylidene difluoride) membrane at 100 mA for 2 h using a Mini-PROTEAN Tetra system (Bio-Rad, USA). For detection of *PpSerpin-1* isoform proteins or PrPAP1, antibodies against PrPAP1 and PpSerpin-1O [22] (diluted 1:1000) were used as primary antibodies, followed by HRP (horseradish peroxidase)-conjugated goat anti-rabbit IgG antibody (Sigma Aldrich, Germany; diluted 1:5000) as the secondary antibody. For detection of His-tagged proteins, THE His Tag mouse antibody (GenScript, Nanjin, China; diluted 1: 2000) was used as the primary antibody, followed by goat anti-mouse IgG antibody-HRP conjugate (Sigma Aldrich, Germany; diluted 1: 5000) as the secondary antibody. The membranes were detected using Pierce ECL Western Blotting Substrate ECL (Thermo Fisher, USA) and imaged by the Chemi Doc-It 600 Imaging System (UVP, Cambridge, UK).

## Recombinant protein expression and purification

For recombinant expression of PpSerpin-1 isoforms, constitutive fragments without signal peptides and alternative fragments were separately amplified using TransTaq HiFi DNA Polymerase (TransGen, Beijing, China) and cloned into the pET-28a vector using the ClonExpress MultiS One Step Cloning Kit (Vazyme, Nanjing, China). Primers were designed using PerlPrimer V1.1.21 and are listed in S2 Table. The linear vector pET-28a was generated by digestion with BamHI and XhoI (TaKaRa, Dalian, China). Recombinant plasmids were then transfected into *E. coli* BL21(DE3) Chemically Competent Cell (TransGen Biotech, Beijing, China) and confirmed by Sanger sequencing (Sangon Biotech, Shanghai, China). *E. coli* cells were grown in autoinduction medium [80] containing 100 µg/µl kanamycin at 300 rpm and 20°C for 48 h and then harvested by centrifugation at 12000 × g for 20 min. Recombinant protein was extracted using BugBuster Protein Extraction Reagent (Thermo Scientific, USA) and purified using cOmplete His-Tag Purification Resin (Roche, Switzerland) and His-Bind Purification kit (Novagen, USA) according to the manufacturer's protocol. The concentration of purified protein was determined using a Pierce BCA Protein Assay Kit (Thermo Scientific, USA).

## Phenoloxidase activity assay

Plasma was harvested by cutting the hind legs of *P. rapae* larvae with scissors and diluted four times into ice-cold TBS buffer (20 mM Tris, 150 mM NaCl, pH = 7.6). Cell-free hemolymph was obtained by centrifugation at 4°C and 3000 × g for 10 min to remove hemocytes. To screen for the inhibitory activities of PpSerpin-1 isoforms on host melanization, 5 µl of recombinant protein (0.2 µg/µl) was mixed with 10 µl of diluted *P. rapae* hemolymph in a 384-well plate. For each sample, 5 µl of elicitor (0.1 µg/µl *M. luteus*) and 5 µl of substrate solution (50 mM L-Dopa in PBS, pH = 7.5) were first mixed and added to another 384-well plate, which was fixed upside down on the sample plate. By centrifuging these two oppositely fixed 384-well plates, the PPO (prophenoloxidase) cascade was activated in each well simultaneously. Plates were measured at A470 and 25°C every 5 min for 2 h using a Varioskan Flash multimode reader (Thermo Scientific, USA). For dose-dependent assays, 5 µl of recombinant protein was mixed with 15 µl of diluted *P. rapae* hemolymph and 5 µl of elicitor (0.1 µg/µl *M. luteus*). After incubation at 25°C for ~10 min, 800 µl of substrate solution (20 mM Dopa in PBS, pH = 6.5) was added. Samples (200 µl) were monitored at A470 in 96-well plates for 20 min using a Varioskan Flash multimode reader (Thermo Scientific, USA). One unit of PO activity was defined as 0.001 △A470/min.

## PrPAP1 amidase activity assay

Recombinant PrPAP1Xa was secretively expressed using the Bac-to-Bac Baculovirus Expression System (Invitrogen, USA) as previously described [22]. Unexpectedly, PrPAP1Xa was activated for unknown reasons. For inhibitory assays, recombinant proteins of PpSerpin-1 isoforms were incubated with activated PrPAP1 at room temperature for 10 min. After adding 200 μl of 50 μM acetyl-Ile-Glu-Ala-Arg-p-nitroanilide (IEARpNA) in TBS (100 mM Tris, 100 mM NaCl, 5 mM CaCl$_2$, pH = 8.0), residual amidase activities were measured at A405 for 20 min using a Varioskan Flash multimode reader (Thermo Scientific, USA). One unit of amidase activity was defined as 0.00001 △A405/min.

## Pull-down assay

Recombinant protein of PpSerpin-1 isoforms (10 μg) was mixed with 1 ml diluted *P. rapae* hemolymph, 50 μl saturated PTU and 100 μl *M. luteus* (1 μg/μl) and incubated on a rotator overnight at 4˚C. After centrifugation at 12000 g and 4˚C for 20 min, the supernatant was further incubated with 25 μl of cOmplete His-tag purification resins (Roche, Switzerland) at 4˚C for 2 h. After washing three times with 300 μl of washing buffer (1 M NaCl, 120 mM imidazole, 40 mM Tris-HCl, pH 7.9), proteins were eluted with 50 μl of elution buffer (1 M imidazole, 0.5 M NaCl, 20 mM Tris-HCl, pH 7.9) and subjected to SDS–PAGE followed by Lumitein Protein Gel Stain (Biotium, Hayward, CA, USA) and immunoblotting. To identify potential targets in pull-down complexes, gels were cut between 60 and 100 kDa. Gel slices were then in-gel digested by trypsin at 37˚C for 20 h. After desalting and lyophilization, the enzymatic product was redissolved in 0.1% formic acid solution and subjected to LC–MS/MS. The parameters used were the same as for the above, except specifically mentioned. The gradient was 1 h. Annotated proteins from *Pieris rapae* genome assembly GCF_001856805.1 were set as the search database. "Oxidation (M)" was set as a variable modification. Protein identification was filtered by proteins with at least two peptides identified. Gel digestion and proteomic identification were conducted by Shanghai Applied Protein Technology Co., Ltd. (Shanghai, China).

## Statistics

All statistical analyses were performed in R v4.1.2. Differences between Spearman correlations were tested using the Hittner2003 method in the R package "cocor" v1.1.4 [81]. To avoid biases that may arise from data non-independence due to shared evolutionary history, phylogenetic correction was performed using the independent contrasts method. Phylogenetic correction allows us to account for the evolutionary relationships among species and obtain more accurate estimates of trait correlations. For independent contrasts, the phylogenetic species tree was generated by phyloT v2 (https://phylot.biobyte.de/). The branch length was estimated by compute.brlen in the R package "ape" v5.6.2 [82] using Grafen's (1989) methods [83]. Independent contrasts were conducted using the "pic" function in the R package "ape" [82]. Correlations through origins were estimated for independent contrasts using the R package "picante" v1.8.2 [84].

## Supporting information

**S1 Data. Excel spreadsheet containing the underlying numerical data for Figs 1–6 in separate sheets.**
(XLSX)

**S1 Table. Primers used for isoform-specific RT-PCR.**
(XLSX)

**S2 Table. Primers used for recombinant expression vector construction.**
(XLSX)

**S3 Table. Proteomic identified proteins in pull-down gel slices.**
(XLSX)

**S1 Fig. Protein and codon alignments at the C-terminal alternative splicing site in the *PpSerpin-1* gene.** The black arrow indicates the splicing site. Curly brackets indicate the hinge region.
(TIF)

**S2 Fig. Comparison of multiple-domain serpin between parasitoid wasps and non-parasitoid hymenopterans.** SPN, serpin.
(TIF)

**S3 Fig. Comparison of gene and exon expansion between parasitoid wasps and non-parasitoid hymenopterans.** Expansion levels were compared using a ranking method. 0 indicates gene-specific expansion; 1 indicates genus-specific expansion; 2 indicates family-specific expansion; 3 indicates order-specific expansion; 4 indicates not specific. Smaller rank numbers mean younger expansions.
(TIF)

**S4 Fig. Comparison of dN/dS between shared and alternative exons.** Constitutive exons include exons 2–7, and alternative exons include 8B, 8C, 8D, 8E, 8F, 8G, 8H, 8I, 8J, 8K, 8X4, 8X5, 8L, 8N, 8O, and 8P.
(TIF)

**S5 Fig. PpSerpin-1 protein secretion into PBS by *P. puparum* larvae.** (a) Western blot detection of PpSerpin-1 proteins in PBS incubated with *P. puparum* larvae. (b) Fluorescence intensity of larvae after 8 h of incubation in PBS. CellTox is a dye that detects cell integrity and can enter ruptured cells and bind to DNA to emit fluorescence. The stronger the fluorescence intensity, the more ruptured cells in the larvae. The larvae remained viable after incubation 8 h, and there was no significant difference compared to the spontaneous fluorescence without CellTox added. Larvae were heat-killed at 100˚C for 10 min and used as the positive control. (c) Representative fluorescence images of *P. puparum* larvae after 8 h of incubation in PBS. * $p < 0.05$; ns.: $p > 0.05$.
(TIF)

**S6 Fig. SDS–PAGE analyses of purified recombinant PpSerpin-1 isoform proteins.** Some isoform proteins show two bands, which might be truncated by-products during the expression process.
(TIF)

**S7 Fig. Dose-dependent suppression of host PO activity by PpSerpin-1A, G, O and P isoform proteins.**
(TIF)

**S8 Fig. Phylogenetic tree of PrHPs in pull-down samples of PpSerpin-1 isoform proteins.** Sequences of MsHP and MsPAP were retrieved from [49] paper. Sequence, alignment and tree files can be downloaded at FigShare (https://doi.org/10.6084/m9.figshare.21545598.v1).
(TIF)

**S9 Fig. Interaction of *Pr*PAP1 with eGFP, G and O.** Black arrows indicate complexes formed for PpSerpin1 isoforms with PrPAP1.
(TIF)

## Acknowledgments

We thank Dr. John H. Werren (University of Rochester, USA) for providing *N. vitripennis*, *T. sarcophagae*, *M. raptor* and *M. uniraptor* species, as well as supervising Z.C.Y. in sampling their saliva. We also thank John H. Werren for his comments and suggestions on the manuscript.

## Author Contributions

**Conceptualization:** Zhichao Yan, Qi Fang, Gongyin Ye.

**Data curation:** Zhichao Yan, Qi Fang.

**Formal analysis:** Zhichao Yan, Qi Fang, Jiqiang Song, Lei Yang, Shan Xiao, Jiale Wang.

**Funding acquisition:** Zhichao Yan, Qi Fang, Lei Yang, Gongyin Ye.

**Investigation:** Zhichao Yan, Qi Fang, Jiqiang Song, Lei Yang, Shan Xiao, Jiale Wang.

**Methodology:** Zhichao Yan, Jiqiang Song, Lei Yang, Gongyin Ye.

**Project administration:** Gongyin Ye.

**Resources:** Gongyin Ye.

**Software:** Zhichao Yan, Qi Fang.

**Supervision:** Gongyin Ye.

**Validation:** Qi Fang, Gongyin Ye.

**Visualization:** Zhichao Yan, Jiqiang Song.

**Writing – original draft:** Zhichao Yan.

**Writing – review & editing:** Gongyin Ye.

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
