## [Decision Letter · Decision Letter 0]

12 Jun 2023

Dear Dr. Ye,

Thank you very much for submitting your manuscript "A parasitoid serpin gene that disrupts host immunity shows adaptive evolution of alternative splicing" for consideration at PLOS Pathogens. As with all papers reviewed by the journal, your manuscript was reviewed by members of the editorial board and by several independent reviewers. In light of the reviews (below this email), we would like to invite the resubmission of a revised version that takes into account the reviewers' comments.The manuscript has received three sets of reviews. We agree with the reviewers that there is a substantial amount of work presented, and the results are novel and advance the field. The reviewers have identified a number of issues to address and these mostly seem fairly straightforward.

We cannot make any decision about publication until we have seen the revised manuscript and your response to the reviewers' comments. Your revised manuscript may be sent to reviewers for further evaluation.

Sincerely,

Francis Michael Jiggins

Academic Editor

PLOS Pathogens

Debra Bessen

Section Editor

PLOS Pathogens

Kasturi Haldar

Editor-in-Chief

PLOS Pathogens

orcid.org/0000-0001-5065-158X

Michael Malim

Editor-in-Chief

PLOS Pathogens

orcid.org/0000-0002-7699-2064

The manuscript has received three sets of reviews. I agree with the reviewers that there is a substantial amount of work presented, and the results are novel and advance the field. The reviewers have identified a number of issues to address and these mostly seem fairly straightforward.

Reviewer's Responses to Questions

**Part I - Summary**

Reviewer #1: This manuscript found a parasitoid serpin gene that disrupts host immunity and shows evolution of alternative splicing (AS). The AS of serpin was firstly reported in Manduca sexta by Jiang et al, 1994 JBC. It will generate up to 12 isoforms in their reactive site loops, which needs to be cited. Here, the authors reported a negative correlation between AS and gene duplication (GD) in the evolution of insect serpin genes and their associations to the parasitic life strategy. The authors also demonstrated how PpSerpin1 isoforms synergistically inhibit the host's immunity in both wasp venom and larval saliva. This is a comprehensive study encompassing evolution and function. GD is a common mechanism for generating new genes, e.g., host effector genes, but there is relatively limited research on the interplay between AS and GD. To my knowledge, this work is also the first functional analysis of a larval salivary gene in parasitoid wasps. Thus, I believe this work represents a significant advancement in understanding the evolution and function of host effector genes in parasitoid wasps.

Overall, I found the manuscript quite interesting. The analyses appear to be detailed, and the conclusions are well supported. I think the manuscript will provide a useful contribution to the field of biointeractions.

Reviewer #2: The manuscript by Yan et al. describes the characterization of the contribution of alternative splicing (AS) to the functional diversity of a serine protease inhibitor (PpSerpin-1) from the venom of a parasitoid wasp (Pteromalus puparum). This work combines a plethora of methods such as genomic and phylogenetic analyses, recombinant protein expression and purification, pull-down assays and proteomics to describe an interesting biological phenomenon. I have no doubt that the extraordinary diversity of splice variants found for PpSerpin-1 was shaped by strong natural selection. However, this manuscript has several issues that should be thoroughly addressed by the authors before I can recommend its acceptance to a scientific journal. Importantly, none of these issues is glaring by itself, but altogether they justify in my opinion a major revision of the manuscript.

Reviewer #3: Although alternative splicing (AS) is considered a major source of protein diversity and evolution of gene expression in eukaryotes (the authors cite 10 reference articles), less is known about its evolution compared to gene duplication (GD) and how alternative gene splicing and gene duplication interact. This is however fundamental to assessing their respective roles in proteic diversity. The authors were original in tackling this complex issue by identifying a very good model for study (strength).

They focused on the evolution of a gene encoding a serpin (PpSerpin1) and compared it between parasitoid wasps and other insects. Using this specific model, they could demonstrate a joint role for alternative splicing and gene duplication in the level of serpins'diversity. This aspect is truly innovative and important in terms of its significance in the evolution of protein diversity mechanisms and ecological adaptation. The study also shows an exemple of how a serpin gene adapts to parasitism through alternative splicing in association with the parasitic life strategy. The knowledge of the literature is very good in terms of alternative splicing, the serpin family and parasitoid virulence factors.

I did not find any problems with the methodology or execution of either the bioinformatics or biochemical aspects. The methods are well described even if some aspects are missing for certain experiments.

For phylogeny, the authors explain that the negative correlations observed hold after phylogenetic correction. It is a pity that this correction is not better explained. It might be useful to explain further the advantages of the AS gene set and the non-AS gene set and the better conservation observed in the AS gene set. For the regulatory sequences, we have 2 Dif and Dorsal motifs enriched in the AS gene set. It would be useful to explain the link with C-term AS. PS in parasitoids compared to non parasitoids Hymenoptera and higher dN/dS values suggesting positive selection. Ppserpin1 was shown to be involved in the Toll pathway and retrieved in salivary gland and venom.

**Part II – Major Issues: Key Experiments Required for Acceptance**

Reviewer #1: PpSerpin1 protein was detected by Western blot in PBS incubated with P. puparum larvae and in larval saliva of P. puparum, N. vitripennis, T. sarcophagae, M. raptor, and M. uniraptor. Could the presence of PpSerpin1 protein be a result of contamination from the rupture of insect bodies? In Figure 4d and 4e, there are bands above 55 kDa, suggesting the existence of complexes formed by PpSerpin1 and its target proteases. Could these complexes be formed between PpSerpin1 and the proteases released from the parasitoid or host larvae?

Reviewer #2: Major comments:

1. The title is very problematic. The “parasitoid serpin gene” is a very confusing term. Also what does “showing adaptive evolution” mean? How about writing instead “A gene encoding a serine protease inhibitor from a parasitoid wasp disrupts host immunity and exhibits adaptive alternative splicing”? If this is too long, think about shorter alternatives that are less confusing and more informative than the current title.

2. While the role of PpSerpin-1 in the venom is demonstrated, it is unclear to me whether any role in the saliva of the larva is demonstrated. Is the assumption that the larva releases via saliva the serpin that inhibits host melanization? This needs to be clearly explained in the text.

3. Lines 253-271: it is unclear how the proteomic analysis contributed to the understanding of the role of the splice variants that do not cause melanization. Please explain.

4. In many genes not all splice variants are functional because AS is a somewhat stochastic process that introduces some “noise”. This possibility should be mentioned in the discussion. Please see Wan and Larson 2018 Genome Biology 19: 86.

5. It seems that the proteomics raw data was not submitted to a public database. The LC-MS/MS data must be submitted to a public database such as PRIDE by EMBL-EBI (see https://www.ebi.ac.uk/pride/) and be publicly available.

Reviewer #3: i) In computer analyses (here, phylogeny), it would be a good idea to include a paragraph explaining why it is important to make a phylogenetic correction (avoid biais) and detailing the methodology used to do so.

ii) I think it would be interesting to go into more detail in the motif scanning approach in the manuscript, including the precise methodology which is not yet available.

iii) The most conclusive experiment is clearly based on the production of PpSerpin 1 isoforms (constitutive fragments without signal peptides), a production that makes it possible to test their capacity to inhibit host phenoloxidase activity, including a dose-dependent aspect. It would have been interesting in this context to test the association of some of these isoforms. The pull-down assay experiments carried out to identify potential interacting targets could have been extended to more isoforms, but it would have been time-consuming and the results are already very conclusive with the identification of target host serine proteases and complexes.

**Part III – Minor Issues: Editorial and Data Presentation Modifications**

Reviewer #1: (1) As shown in Fig. 4e, the presence of the PpSerpin1 protein seems to be widespread in the larval saliva of P. puparum and its relatives. Accordingly, it is worth investigating whether the PpSerpin1 protein is also found in the venom of these parasitoid species. This can be explored by Western blotting utilizing PpSerpin1 antibodies. Furthermore, several published articles have addressed the identification of venom proteins in Nasonia and its relatives. For instance, Martinson et al. (2017) reported the venom proteome of N. vitripennis, N. giraulti, T. sarcophagae, and U. rufipes. It would be interesting to determine if the PpSerpin1 protein is also present in the venom proteomic data reported in these studies.

Reference: Martinson EO, Mrinalini, Kelkar YD, Chang CH, Werren JH. The evolution of venom by co-option of single-copy genes. Curr Biol. 2017;27(13):2007-13.

(2) The methods used for immune simulation must be clearly described. A related question: Is there any overlap between the isoforms induced by immune stimulation and the isoforms recruited to venom or larval saliva? Is there a correlation between the expression levels of isoforms in the venom gland or larval salivary gland and the fold changes after immune simulation? Are the isoforms that are upregulated by immune stimulation more likely to be recruited into venom or larval saliva?

(3) The authors compared several features between parasitoid wasps and non-parasitoid insects. In addition to parasitoid wasps, are there any other parasitic insects included in the analyzed dataset? If so, do the findings regarding parasitoid wasps also apply to these other parasitic insects?

(4) For the protein structure prediction, it is necessary to provide more details and parameters, including whether templates were used in AlphaFold2 and the confidence level of predicted models, such as pLDDT values.

(5) In abstract, “…have been recruited to both wasp venom and larval saliva”. Again, is there any evidence, such as Wb or proteomics (apart from Fig 4c), to show that Ppserpin in the venom? Probably it is related to the previous publication.

(6) In Figure 6, “Pp” in “PpSerpin-1” is italicized, while “Serpin-1” is in regular font. Similarly, “Pr” in “PrPAP1” is italicized, while “PAP1” is in regular font. Generally, gene names should be italicized, while protein names should not be. It is important to ensure consistency throughout the entire document in adhering to these conventions.

(7) The clarity of Fig 1a and 1c needs to be improved. The lines in the figures are not clear, which hinders readability.

(8) In line 37, “Pteromalus puparum” should be italicized.

(9) In line 91, change “alternate” to “alternative”

(10) In line 913, change “Western blot detection in larval saliva of Nasonia vitripennis, Trichomalopsis sarcophagae, Muscidifurax raptor and M. uniraptor using PpSerpin1 antibodies.” to “Western blot detection in larval saliva of P. puparum, Nasonia vitripennis, Trichomalopsis sarcophagae, Muscidifurax raptor, and M. uniraptor using PpSerpin1 antibodies.”

Reviewer #2: Minor comments:

1. Lines 47-48: “ubiquitous” is a confusing term in the context of this sentence. Please change to “frequent”.

2. Lines 221-223: the term “accelerated protein evolution” is a bit outdated. Please use the term “positive selection diversifying protein sequence” instead.

3. dN and dS should be italicized.

4. Lines 230-231: this suggestion is highly speculative and should be moved to the discussion. Furthermore, it should be emphasized this is based on data from Drosophila and not from wasps.

5. Line 517: gastropod is a mollusk. I think you mean another term?

6. Lines 907-908: the figure legend of panel 4b should be more detailed. I guess this heatmap represents RNA-Seq results? This should be clarified.

7. Lines 919-920: why are there two bands for some of the recombinant proteins? Is this incomplete denaturation or some truncated by-product?

Reviewer #3: i) In some places in the manuscript, certain words are misspelt (letters not in the right order) and should be corrected.

ii) I find that the figures are a bit crowded (number of graphs per figure) and I suggest simplifying them where possible and desirable.

PLOS authors have the option to publish the peer review history of their article (what does this mean?). If published, this will include your full peer review and any attached files.

Reviewer #1: No

Reviewer #2: No

Reviewer #3: No
---

## [Decision Letter · Decision Letter 1]

31 Aug 2023

Dear Dr. Ye,

We are pleased to inform you that your manuscript 'A serpin gene from a parasitoid wasp disrupts host immunity and exhibits adaptive alternative splicing' has been provisionally accepted for publication in PLOS Pathogens.

Best regards,

Francis Michael Jiggins

Academic Editor

PLOS Pathogens

Debra Bessen

Section Editor

PLOS Pathogens

Kasturi Haldar

Editor-in-Chief

PLOS Pathogens

orcid.org/0000-0001-5065-158X

Michael Malim

Editor-in-Chief

PLOS Pathogens

orcid.org/0000-0002-7699-2064

Reviewer Comments (if any, and for reference):

Reviewer's Responses to Questions

**Part I - Summary**

Reviewer #1: The authers have answered all my concerns, and I have no more questions.

Reviewer #2: (No Response)

Reviewer #3: I have read the revised version of Dr Gongyin Ye's article with great attention and I am satisfied with the modifications and additions made to the first version in response to my questions and suggestions. The changes made in response to the comments of the other 2 reviewers also seem to me to be entirely relevant and to bring clarification to the document. Finally, I'd like to highlight the work carried out to produce additional experiments to support the results obtained and the conclusions drawn.

As a result, I believe that this article has all its place in PLoS Pathogens and should be accepted.

**Part II – Major Issues: Key Experiments Required for Acceptance**

Reviewer #1: (No Response)

Reviewer #2: There are no remaining major or minor issues. The authors have adequately addressed all my previous comments.

Reviewer #3: (No Response)

**Part III – Minor Issues: Editorial and Data Presentation Modifications**

Reviewer #1: (No Response)

Reviewer #2: (No Response)

Reviewer #3: (No Response)

PLOS authors have the option to publish the peer review history of their article (what does this mean?). If published, this will include your full peer review and any attached files.

Reviewer #1: No

Reviewer #2: No

Reviewer #3: No

---

## [Editor Report · Acceptance letter]

6 Sep 2023

Dear Dr. Ye,

We are delighted to inform you that your manuscript, "A serpin gene from a parasitoid wasp disrupts host immunity and exhibits adaptive alternative splicing," has been formally accepted for publication in PLOS Pathogens.

Best regards,

Kasturi Haldar

Editor-in-Chief

PLOS Pathogens

orcid.org/0000-0001-5065-158X

Michael Malim

Editor-in-Chief

PLOS Pathogens

orcid.org/0000-0002-7699-2064